# Context-level Language Modeling by Learning Predictive Context Embeddings

**Beiya Dai** [1]   **Yuliang Liu** [1 5]   **Yunchong Song** [2]   **Daozheng Xue** [1]   **Qipeng Guo** [2]   **Kai Chen** [2]   **Xinbing Wang** [3]
**Bowen Zhou** [2 4]   **Zhouhan Lin** [1 2]

## Abstract

We propose ContextLM, a framework that implicitly learns multi-token prediction by augmenting standard pretraining with an intrinsic next-context prediction objective. ContextLM builds a language model on top of context embeddings that span multiple tokens, enabling better next-token prediction by predicting the next context. Our model is fully compatible with standard autoregressive, token-by-token evaluation paradigms (e.g., perplexity). Extensive experiments with GPT-2 and Pythia backbones (up to 1.5B parameters and 300B training tokens) reveal that ContextLM shifts the Pareto frontier of scaling laws, exhibiting superior efficiency in parameters, training tokens, and FLOPs. Our results show that ContextLM could already achieve the baseline perplexity using 39% fewer parameters and demonstrates robust generalization improvements on extensive downstream tasks under equivalent parameter counts.

## 1. Introduction

Large language models (LLMs) have become the foundation of modern natural language processing (NLP), demonstrating remarkable capabilities in text generation, logical reasoning, and generalization ability (Radford et al., 2019; Brown et al., 2020; Achiam et al., 2023; Touvron et al., 2023). Historically, language models have mainly been explored at two levels of granularity: character-level (Graves, 2013; Kim et al., 2016) and token-level (Bengio et al., 2003). Modern LLMs have universally converged on the token-level paradigm, as token-level prediction provides a more challenging training objective. This dominance of token-level models stems from its requirement to predict over more so-

phisticated dynamics that transcend character combinations, thereby yielding more powerful models.

Recent works have explored more challenging pretraining tasks, such as multi-token prediction (MTP), which extends next-token prediction (NTP) to predict multiple future tokens (Gloeckle et al., 2024; Shao et al., 2024). Although MTP advances beyond token-level prediction via its training objective, the model's hidden representations remain at the token level. Moving beyond such token-level representations requires fundamental architectural innovations. The pivotal challenge here lies in the construction and utilization of abstract-level representations. The pioneering framework JEPA (LeCun, 2022) addresses this by introducing a systematic approach to learning multi-level representations, shifting the predictive objective to a latent representation space. This framework has been successfully applied in the image and video domains (Bardes et al., 2023; Tuncay et al., 2025).

For natural language, efforts to introduce hierarchical abstraction have been made primarily through two directions. Sentence-level abstractions like LCM (Barrault et al., 2024) and Block Transformer (Ho et al., 2024) successfully compress information at the cost of sacrificing autoregressive modeling in the latent space. On the other hand, character-level methods such as BLT (Pagnoni et al., 2025) and H-Net (Hwang et al., 2025) construct latent token-level representations by dynamically grouping characters into patches. While these models can learn to form dynamic token-level representations through training, their effectiveness to learn representations beyond the token-level still remains challenging.

In this work, we propose **ContextLM**, a framework that introduces an extra intrinsic context-level prediction mechanism that operates in the latent space. Our approach decomposes the Transformer into a Token Encoder and a Token Decoder, with an additional Context Predictor in between. This predictor operates autoregressively in latent space, generating predictive context embeddings to semantically guide the token generation. Instead of explicitly predicting discrete tokens, ContextLM employs an intrinsic next-context prediction objective for the Context Predictor, optimizing contexts to minimize aggregated signals from future tokens.

[1]LUMIA Lab, Shanghai Jiao Tong University [2]Shanghai AI Laboratory [3]Shanghai Jiao Tong University [4]Tsinghua University [5]Nanjing University. Correspondence to: Zhouhan Lin <lin.zhouhan@gmail.com>.

*Proceedings of the $43^{rd}$ International Conference on Machine Learning*, Seoul, South Korea. PMLR 306, 2026. Copyright 2026 by the author(s).

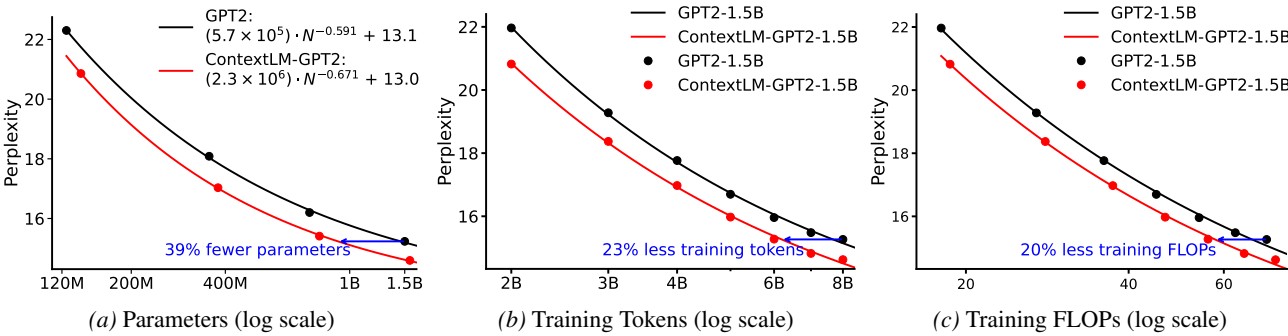

*Figure 1.* Scaling performance comparison across three dimensions: parameters, training tokens, and training FLOPs for GPT-2 and ContextLM-GPT2.

Crucially, ContextLM maintains full architectural compatibility with standard Transformers, allowing seamless integration with existing mainstream LLM evaluation metrics such as token-level perplexity.

Our empirical evaluation demonstrates that ContextLM significantly shifts the Pareto frontier of efficient language modeling. Using GPT-2 backbones, ContextLM matches baseline perplexity using 39% fewer parameters, 23% fewer training tokens, and 20% fewer FLOPs (see Figure 1). We further validate the scalability of our approach by training a ContextLM variant based on the Pythia-1.4B architecture on 300B tokens. The results show that ContextLM not only reduces perplexity but also translates these gains into superior downstream task performance and stronger instruction-following capabilities, suggesting that latent context prediction offers a promising path toward stronger language models.

## 2. ContextLM

In this section, we present ContextLM, which constructs predictive context embeddings optimized by aggregated multi-token error signals, thereby augmenting conventional next-token modeling. We first introduce the problem setup, followed by the model architecture and training objective. In addition, we provide an analysis of the computational complexity in Appendix A.1.1.

### 2.1. Problem Setup

Given a text corpus $\mathcal{C}$, the standard language modeling objective is to model sequence $\mathbf{x}_{0:T-1} = (x_0, x_1, \ldots, x_{T-1})$, where $\mathbf{x}_{0:T-1} \in \mathcal{C}$, using a unidirectional autoregressive model $\theta_0$. This standard language model estimates token probabilities $p_{\theta_0}(x_t \mid x_{<t})$ based solely on preceding tokens.

We aim to augment the standard language model with latent, context-level representations that encode semantic dependencies beyond adjacent tokens. To this end, we introduce

a hierarchical architecture comprising three components: a Token Encoder to extract token-level representations, a Context Predictor to model the evolution of high-level semantics across chunks, and a Token Decoder to generate text conditioned on both token-level and predicted context embeddings.

The Context Predictor is trained to autoregressively predict the embedding of the next context. The resulting model defines a context-aware language model $\theta$ that conditions token generation on both previous tokens and predicted context embeddings: $\theta(x_t \mid x_{<t}, \hat{c}_k)$, where $k = \lfloor t/w \rfloor + 1$ denotes the current chunk index and $w$ is the chunk size. Here, $\hat{c}_k$ is the predicted context embedding corresponding to the current chunk.

### 2.2. Model Architecture

The overall architecture of ContextLM, illustrated in Figure 2, extends standard NTP with a context prediction pathway to capture multi-granularity semantic dependencies. It consists of three principal components: Token Encoder, Context Predictor, and Token Decoder.

#### 2.2.1. TOKEN ENCODER $\mathcal{E}$

The Token Encoder $\mathcal{E}$ follows a standard decoder-only Transformer design and operates at the token level. Given an input token sequence $\mathbf{x}_{0:T-1}$, it produces token-level embeddings:

$$\mathbf{h}_{0:T-1} = \mathcal{E}(x_{0:T-1}). \tag{1}$$

As shown in Figure 2 (bottom), the token sequence is divided into fixed-length chunks of size $w$. These token embeddings serve two purposes: (i) providing fine-grained token representations for NTP, and (ii) supplying the foundation for constructing higher-level context embeddings.

#### 2.2.2. CONTEXT PREDICTOR $\mathcal{P}$

The Context Predictor $\mathcal{P}$ operates on a sequence of context embeddings derived from token representations. A mapping function $f(\cdot)$ aggregates token-level hidden states within

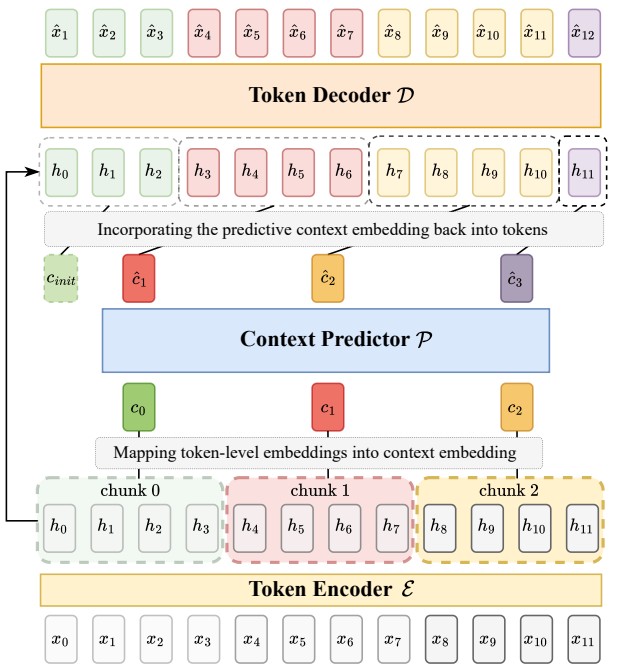

**Algorithm 1: ContextLM Inference**

**Require:** $\mathbf{x}_{0:T-1}$, $w$, $L$, $T$, $K = \lfloor T/w \rfloor + 1$
**Ensure:** $\mathbf{x}_{0:T+L-1}$
    *# Initialize embeddings*
1:   $\mathbf{h}_{0:T-1} \leftarrow \mathcal{E}(\mathbf{x}_{0:T-1})$
2:   $\mathbf{c}_{0:K-1} \leftarrow f(\mathbf{h}_{0:T-1})$
3:   $\hat{\mathbf{c}}_K \leftarrow \mathcal{P}(\mathbf{c}_{0:K-1})$
4: **for** $t = 0$ **to** $L - 1$ **do**
      *# Predict next token*
5:      $\mathbf{p} \leftarrow \text{Softmax}(\mathcal{D}(\mathbf{h}_{0:T+t-1}, \hat{\mathbf{c}}_K))$
6:      $x_{T+t} \sim \mathbf{p}$
7:      $\mathbf{h}_{0:T+t} \leftarrow [\mathbf{h}_{0:T+t-1}, \mathcal{E}(x_{T+t})]$
8:      **if** $(T + t + 1) \bmod w = 0$ **then**
9:          $\mathbf{c}_K \leftarrow f(\mathbf{h}_{Kw:(K+1)w})$
10:        $\mathbf{c}_{0:K} \leftarrow [\mathbf{c}_{0:K-1}, \mathbf{c}_K]$
11:        $K \leftarrow K + 1$
        *# Predict next context*
12:       $\hat{\mathbf{c}}_K \leftarrow \mathcal{P}(\mathbf{c}_{0:K-1})$
13:     **end if**
14: **end for**
15: **return** $\mathbf{x}_{0:T+L-1}$

*Figure 2.* Overview of the ContextLM architecture. The model extends the standard language model by introducing a Context Predictor $\mathcal{P}$ which generates context embeddings that are fed into the Token Decoder $\mathcal{D}$ to guide token generation. The pseudocode details the inference procedure, showing the integration of context prediction into the token generation loop. Notably, the context prediction step only occurs at chunk boundaries.

each chunk:

$$\mathbf{c}_{0:K-1} = f(\mathbf{h}_{0:T-1}), \quad K = \lfloor T/w \rfloor + 1. \quad (2)$$

The Context Predictor then performs autoregressive modeling over this context sequence:

$$\hat{\mathbf{c}}_{1:K} = \mathcal{P}(\mathbf{c}_{0:K-1}), \quad (3)$$

where each $\hat{c}_k$ depends only on previously inferred contexts.

To handle boundary cases where the first chunk has no preceding context available, we introduce an initialization embedding $c_{init}$, which is prepended to the context sequence and serves as a prior context embedding before the first prediction:

$$\hat{\mathbf{c}} = (c_{init}, \hat{c}_1, \ldots, \hat{c}_K). \quad (4)$$

When fed $c_{init}$ into the Token Decoder, $c_{init}$ serves as the null hypothesis for the context information in the first chunk of tokens.

Notably, context prediction occurs only at chunk boundaries (see Algorithm 1), leading to substantially fewer prediction steps than token-level modeling.

### 2.2.3. TOKEN DECODER $\mathcal{D}$

The Token Decoder $\mathcal{D}$ performs standard NTP conditioning on both token embeddings and predicting context embeddings. To align granularities, each predicted context $\hat{c}_k$ is

broadcast to all tokens within its corresponding chunk:

$$\hat{\mathbf{c}}^b = (\underbrace{c_{init}, \ldots, c_{init}}_{w-1 \text{ times}}, \underbrace{\hat{c}_1, \ldots, \hat{c}_1}_{w \text{ times}}, \ldots, \underbrace{\hat{c}_{K-1}, \ldots, \hat{c}_{K-1}}_{T+1-(K-1)w \text{ times}}). \quad (5)$$

The decoder then combines multi-level information via element-wise addition $h \oplus \hat{c}^b$. The decoder then proceeds with causal attention and softmax-based token prediction.

All components employ causal attention to strictly enforce autoregressive constraints and prevent information leakage. Temporal causality is guaranteed through consistent one-step lookahead logic at each hierarchical level, as illustrated in Figure 2. At the token level, the Token Decoder follows the standard **one-token shift**: when predicting token $x_t$, the hidden state $h_t$ is conditioned on previously generated tokens $x_{<t}$. At the context level, tokens are partitioned into non-overlapping chunks, with each chunk $k$ summarized into a context embedding $c_k$. The Context Predictor applies a **one-chunk shift**, predicting $\hat{c}_k$ based solely on past context embedding $c_{<k}$. By aligning these two autoregressive processes across token-level and context-level modeling, ContextLM maintains full compatibility with standard NTP while strictly preventing information leakage.

### 2.3. Model Training

ContextLM retains the standard token-level cross-entropy loss $\mathcal{L}_{CE}$, ensuring compatibility with mainstream token-

level models. In addition, unlike standard NTP where logits $z_t = \mathcal{D}(h_t)$ depend solely on local token representations, ContextLM conditions generation on predicted context embeddings: $z'_t = \mathcal{D}(h_t, \hat{c}_k)$. This modification fundamentally reshapes the gradient flow during backpropagation, providing richer multi-level supervision while preserving the original training objective. Specifically:

- **Context-level Supervision for Context Predictor.** Each predicted context embedding $\hat{c}_k$ conditions all tokens within its corresponding chunk set $\mathcal{J}_k$. Consequently, it receives an aggregated error signal summed across these token positions:

$$\frac{\partial \mathcal{L}_{CE}}{\partial \hat{c}_k} = \sum_{j \in \mathcal{J}_k} \frac{\partial \mathcal{L}_{CE}}{\partial z'_j} \frac{\partial z'_j}{\partial \hat{c}_k}. \tag{6}$$

This aggregated supervision encourages the Context Predictor to capture high-level semantics relevant to the entire chunk rather than adapting to individual local patterns.

- **Multi-level Supervision for Token Encoder.** The Token Encoder parameters receive both token-level and context-level gradient signals. The gradient for $h_t$ is composed of the direct local signal and the indirect context signal:

$$\frac{\partial \mathcal{L}_{CE}}{\partial h_t} = \underbrace{\frac{\partial \mathcal{L}_{CE}}{\partial z'_t} \frac{\partial z'_t}{\partial h_t}}_{\text{token-level signal}} + \underbrace{\left( \sum_{j \in \mathcal{J}_k} \frac{\partial \mathcal{L}_{CE}}{\partial z'_j} \frac{\partial z'_j}{\partial \hat{c}_k} \right) \frac{\partial \hat{c}_k}{\partial h_t}}_{\text{context-level signal}}. \tag{7}$$

This ensures that the encoder representations are optimized not only for immediate NTP but also for constructing robust context representations that facilitate multi-step future generation. [1]

## 3. Experiments

We conduct extensive experiments to comprehensively assess the effectiveness, scalability, and generality of ContextLM across different backbones, dataset scales, and evaluation settings. The experiments cover four main aspects: (i) scaling law behavior on both GPT-2 and Pythia families (Sec. 3.2); (ii) comprehensive evaluation on diverse downstream tasks (Sec. 3.3); (iii) instruction-following performance after fine-tuning (Sec. 3.4); and (iv) ablation studies and analysis (Sec. 4).

---

[1]Even when the chunk size is set to $w = 1$, ContextLM does not equal to standard NTP. In this case, each token corresponds to a single context chunk. At step $t$, the model generates $x_{t+1}$ by conditioning on the token representation $h_t$ and a predictive context $\hat{c}_{t+1}$ (inferred from $c_{\leq t}$), resulting in a hierarchical two-step prediction process.

### 3.1. Experimental Setup

**Models and Datasets**  We evaluate ContextLM on two widely used Transformer families to demonstrate its architectural compatibility and effectiveness. (i) **GPT-2**: models are pretrained from scratch on the OpenWebText (Gokaslan & Cohen, 2019) with a maximum sequence length of 1024. Additional implementation details on GPT-2 are provided in Appendix A.2. (ii) **Pythia**: to examine large-scale scalability, we train ContextLM-Pythia from scratch on the entire Pile dataset (Gao et al., 2020) (300B tokens) with a maximum sequence length of 2048, following the original Pythia tokenizer and training configuration (optimizer settings, learning rate schedule, batch size, and context length).

**ContextLM Settings**  In our practice, we use the mean pooling function as the mapping function $f(\cdot)$ in Sec. 2.2. We use the first token embedding $h_0$ as $c_{init}$. For Pythia models that use PoSE as the position embedding technique, we use the position of the first token in each chunk as the context chunk position in context layers. Unless specifically noted, we set the context chunk size to $w = 4$ and use a two-layer Context Predictor.

*Table 1.* Performance comparison ContextLM with baseline on GPT-2 backbone across different model scales. See Appendix A.5 for full task details.

| Model | Avg PPL↓ | Avg Acc ↑ | |
| --- | --- | --- | --- |
| | | 0-shot | 5-shot |
| GPT2-Base-124M | 110.92 | 37.8 | 36.6 |
| GPT2-Base-PM | 107.77 | 38.0 | 36.7 |
| **ContextLM-Base** | **87.19**$_{\downarrow 23.73}$ | **39.1**$_{\uparrow 1.3}$ | **37.6**$_{\uparrow 1.0}$ |
| GPT2-Medium-355M | 55.46 | 40.5 | 38.8 |
| GPT2-Medium-PM | 53.18 | 41.0 | 39.4 |
| **ContextLM-Medium** | **43.06**$_{\downarrow 12.40}$ | **41.9**$_{\uparrow 1.4}$ | **40.2**$_{\uparrow 1.4}$ |
| GPT2-Large-774M | 38.27 | 41.4 | 41.0 |
| GPT2-Large-PM | 36.28 | 42.4 | 41.3 |
| **ContextLM-Large** | **31.80**$_{\downarrow 6.47}$ | **43.9**$_{\uparrow 2.5}$ | **43.1**$_{\uparrow 2.1}$ |
| GPT2-XL-1.5B | 31.54 | 43.3 | 42.1 |
| GPT2-XL-PM | 31.57 | 43.2 | 42.4 |
| **ContextLM-XL** | **25.93**$_{\downarrow 5.61}$ | **45.0**$_{\uparrow 1.7}$ | **44.0**$_{\uparrow 1.9}$ |

### 3.2. Scaling Experiments

**Scaling Law on GPT-2**  We begin by evaluating the scaling behavior of ContextLM on the GPT-2 family, referred to ContextLM-GPT2, trained on OpenWebText under matched compute budgets of the baseline GPT-2. Figure 1 reports the perplexity as a function of model parameters, training tokens, and training FLOPs. Across all three scaling dimensions, our model consistently outperforms the GPT-2

*Table 2.* Perplexity Comparison of Pythia and ContextLM-Pythia across four benchmark datasets. For brevity, we denote ContextLM-Pythia as ContextLM. "Avg PPL" reports the average perplexity per model, where lower values indicate better performance.

| Model | Pile | Wikitext | Lambada OpenAI | Lambada Standard | Avg PPL ↓ |
|---|---|---|---|---|---|
| Pythia-70M | 18.27 | 57.01 | 142.01 | 973.59 | 297.72 |
| **ContextLM-70M** | **14.96**↓ 3.31 | **43.64**↓ 13.37 | **71.45**↓ 70.56 | **440.77**↓ 532.82 | **142.71**↓ 155.01 |
| Pythia-160M | 12.56 | 33.44 | 38.20 | 187.28 | 67.87 |
| **ContextLM-160M** | **11.14**↓ 1.42 | **28.18**↓ 5.26 | **25.97**↓ 12.23 | **107.05**↓ 80.23 | **43.09**↓ 24.78 |
| Pythia-410M | 8.88 | 20.11 | 10.85 | 31.53 | 17.84 |
| **ContextLM-410M** | **8.67**↓ 0.21 | **19.50**↓ 0.61 | **10.15**↓ 0.70 | **24.95**↓ 6.58 | **15.82**↓ 2.02 |
| Pythia-1B | 7.82 | 16.45 | 7.92 | 17.44 | 12.41 |
| **ContextLM-1B** | **7.66**↓ 0.16 | **16.09**↓ 0.36 | **7.38**↓ 0.54 | **15.75**↓ 1.69 | **11.72**↓ 0.69 |
| Pythia-1.4B | 7.26 | 14.72 | 6.09 | 10.87 | 9.74 |
| **ContextLM-1.4B** | **7.16**↓ 0.10 | **14.61**↓ 0.11 | **6.06**↓ 0.03 | **10.30**↓ 0.57 | **9.54**↓ 0.20 |

baseline. Moreover, we compare ContextLM against the Parameter-Matched (PM) baselines, constructed by adding two standard Transformer layers, in Table 1. These results demonstrate the favorable scaling of our method.

**Scaling Law on Pythia** To examine the general applicability and effectiveness of ContextLM, we further evaluate it on the Pythia family (70M–1.4B parameters) to assess scalability across larger data (300B tokens). As shown in Table 2, ContextLM-Pythia consistently achieves lower perplexity across the Pile, Wikitext (Merity et al., 2016), and Lambada (Paperno et al., 2016). The relative improvement is most pronounced at smaller scales and remains consistent as model size increases, confirming that the benefits of context-level supervision persist throughout scaling. These results demonstrate that ContextLM maintains both scalability and computational efficiency on large-scale datasets.

### 3.3. Downstream Task

To evaluate generalization beyond perplexity, we evaluate the zero-shot and five-shot performance using the `lm-evaluation-harness` [2] across 9 widely-used benchmarks, which we categorize into three core capabilities: linguistic understanding (Lambada OpenAI/Standard (Paperno et al., 2016), WinoGrande (Sakaguchi et al., 2019)), commonsense reasoning (ARC-Easy/Challenge (Clark et al., 2018), PIQA (Bisk et al., 2020), HellaSwag (Zellers et al., 2019), SciQ (Welbl et al., 2017)), and complex reasoning (RACE (Lai et al., 2017)), following the evaluation established in (Gu & Dao, 2024; Zeng et al., 2025).

As shown in Table 3, ContextLM-Pythia consistently outperforms the Pythia baseline across all model sizes in both

evaluation settings. Improvements are observed across all task categories, with larger relative gains at smaller scales and more stable absolute gains at larger scales. Notably, the advantages are more pronounced in the five-shot setting, suggesting that context-level supervision yields representations that transfer more effectively under limited in-context supervision. Additional downstream results on ContextLM-GPT2, GPT-2, and PM-GPT2 baselines are reported in Appendix A.5. Across both GPT-2 and Pythia backbones, ContextLM consistently improves downstream performance over baselines, demonstrating that context-level supervision better utilizes available parameters and generalizes across diverse tasks.

### 3.4. Instruction-following Ability

To assess the instruction-following ability of our model, we further fine-tuned ContextLM-Pythia and official Pythia on the Alpaca (Taori et al., 2023) using the same settings (Taori et al., 2023), and evaluated using MT-Bench (Zheng et al., 2023) and AlpacaEval 2.0 (Dubois et al., 2024). As shown in Figure 3, ContextLM-Pythia consistently surpasses Pythia across multiple capability subtasks. At the 1B scale, the average score improves from 1.62 to 1.83, while the 1.4B model shows more substantial gains, increasing from 1.99 to 2.37. On the more modern AlpacaEval 2.0 benchmark (Table 4), ContextLM achieves a 54.48% win rate over the Pythia-1.4B baseline. These results suggest that context-level supervision not only enhances pretraining efficiency but also facilitates stronger instruction-following performance after fine-tuning.

## 4. Analysis

In this section, we conduct a comprehensive analysis to understand how key choices in ContextLM influence model

---

[2]https://github.com/EleutherAI/lm-evaluation-harness

*Table 3.* Zero-shot and five-shot evaluation results across 9 downstream benchmarks. "Avg Acc" denotes the mean accuracy per model across each evaluation condition, with higher values indicating better performance.

| Model | Lambada OpenAI | ARC-E | Lambada Standard | ARC-C | Wino Grande | PIQA | Hella-Swag | SciQ | RACE | Avg Acc ↑ |
|---|---|---|---|---|---|---|---|---|---|---|
| *Zero-shot* | | | | | | | | | | |
| Pythia-70M | 18.3 | 36.9 | 13.4 | 18.5 | 52.1 | 60.0 | 26.6 | 60.5 | 24.9 | 34.6 |
| **ContextLM-70M** | 28.0 | 40.7 | 17.2 | 18.6 | 52.3 | 60.5 | 27.5 | 72.1 | 27.0 | **38.2 / ↑ 3.6** |
| Pythia-160M | 32.7 | 43.8 | 21.5 | 19.5 | 53.4 | 61.5 | 28.5 | 74.3 | 27.9 | 40.3 |
| **ContextLM-160M** | 37.2 | 45.0 | 25.6 | 19.5 | 52.0 | 62.9 | 29.2 | 77.6 | 28.7 | **42.0 / ↑ 1.7** |
| Pythia-410M | 51.6 | 52.2 | 36.4 | 21.3 | 53.9 | 66.8 | 33.8 | 81.2 | 30.7 | 47.5 |
| **ContextLM-410M** | 52.2 | 51.7 | 39.0 | 22.7 | 52.3 | 67.4 | 34.3 | 83.0 | 30.8 | **48.2 / ↑ 0.7** |
| Pythia-1B | 55.9 | 56.8 | 42.0 | 24.2 | 52.5 | 70.5 | 37.7 | 83.3 | 32.7 | 50.6 |
| **ContextLM-1B** | 57.8 | 55.3 | 43.7 | 25.9 | 55.0 | 70.1 | 37.6 | 86.1 | 32.1 | **51.5 / ↑ 0.9** |
| Pythia-1.4B | 61.6 | 60.4 | 49.7 | 25.9 | 57.5 | 70.8 | 40.4 | 86.4 | 34.1 | 54.1 |
| **ContextLM-1.4B** | 61.6 | 58.9 | 51.4 | 27.2 | 55.7 | 71.2 | 40.6 | 87.9 | 34.9 | **54.4 / ↑ 0.3** |
| *Five-shot* | | | | | | | | | | |
| Pythia-70M | 11.9 | 36.7 | 9.2 | 17.1 | 50.5 | 58.7 | 26.7 | 57.8 | 25.1 | 32.6 |
| **ContextLM-70M** | 19.2 | 41.4 | 14.2 | 18.5 | 51.1 | 60.8 | 27.7 | 71.5 | 25.7 | **36.7 / ↑ 4.1** |
| Pythia-160M | 24.9 | 44.7 | 19.0 | 18.4 | 50.4 | 63.5 | 28.6 | 76.4 | 27.8 | 39.3 |
| **ContextLM-160M** | 29.1 | 45.7 | 23.1 | 19.8 | 51.9 | 63.1 | 29.5 | 81.7 | 28.2 | **41.3 / ↑ 2.0** |
| Pythia-410M | 43.9 | 54.7 | 32.8 | 22.3 | 53.4 | 68.0 | 33.8 | 88.9 | 30.4 | 47.6 |
| **ContextLM-410M** | 44.6 | 54.8 | 34.9 | 23.0 | 52.7 | 68.0 | 34.3 | 89.5 | 30.9 | **48.1 / ↑ 0.5** |
| Pythia-1B | 48.3 | 58.6 | 35.8 | 25.4 | 52.8 | 71.3 | 37.7 | 91.6 | 31.7 | 50.4 |
| **ContextLM-1B** | 49.9 | 60.2 | 39.5 | 24.2 | 54.9 | 70.5 | 38.1 | 91.5 | 32.6 | **51.3 / ↑ 0.9** |
| Pythia-1.4B | 54.5 | 63.1 | 44.5 | 28.8 | 57.1 | 71.0 | 40.5 | 92.4 | 34.6 | 54.1 |
| **ContextLM-1.4B** | 55.7 | 62.3 | 46.7 | 28.5 | 56.8 | 72.4 | 41.1 | 93.3 | 35.0 | **54.6 / ↑ 0.5** |

*Table 4.* AlpacaEval 2.0 Win Rate Comparison. Evaluated using Pythia-1.4B-SFT as the reference model.

| Model | Win Rate (%) | LC Win Rate (%) |
|---|---|---|
| Pythia-1.4B | 45.52 | 45.99 |
| **ContextLM-Pythia-1.4B** | **54.48** | **54.01** |

behavior and empirical performance.

### 4.1. Ablation Study

**Chunk Size** We first study the effect of chunk size $w$ by varying it across $\{2, 4, 8, 16\}$. To enable a fair comparison under comparable computational budgets, we adjust the depth of the Context Predictor with respect to $w$, as illustrated in Figure 4 (a). Specifically, as increasing $w$ reduces the effective sequence length processed by the predictor, we increase its depth to keep the overall FLOPs and memory approximately constant. Our results indicate that $w = 4$ offers the best trade-off, so we adopt $w = 4$ as the default setting.

**Context Predictor Depth** We next analyze the effect of the Context Predictor depth (Figure 4 (b)). Increasing the predictor depth beyond two layers results in only marginal gains, indicating that a 2-layer predictor is sufficient to capture the dominant semantic transitions for effective context-level modeling; we therefore adopt a 2-layer Context Predictor as the default configuration.

**Context Injection Layer** We investigate where to inject the Context Predictor by varying the split layer between the Token Encoder and Token Decoder. As shown in Figure 4 (c), the 0/12 setting provides the best performance. This configuration allows more decoder layers to process and integrate the contextual signal, leading to deeper semantic integration. Furthermore, since shallower layers primarily encode local and low-level features, the predictor must therefore infer high-level semantics from limited contextual cues, making the learning task more challenging.

**Block Size** We further examine the effect of block size ranging from 512 to 2048 tokens (Figure 4 (d)). Since

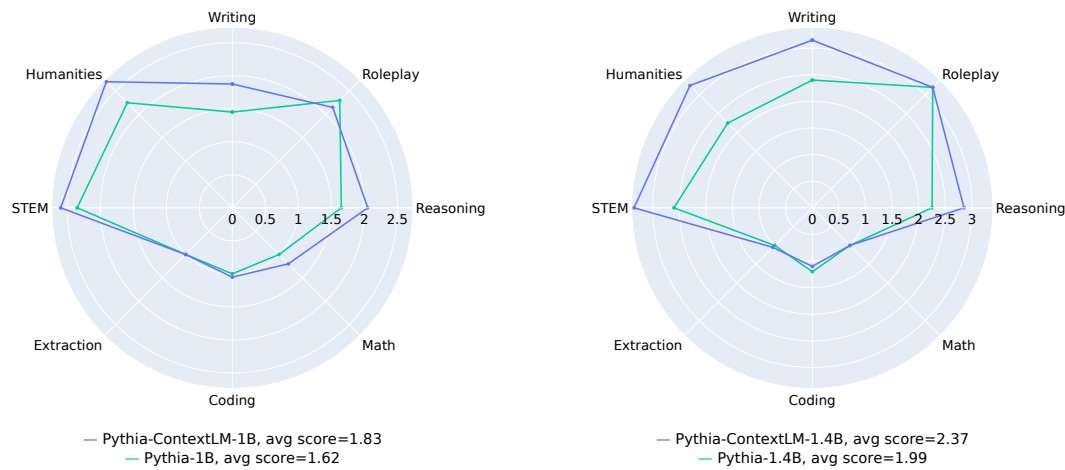

Figure 3. Instruction-following evaluation on MT-Bench across multiple subtasks.

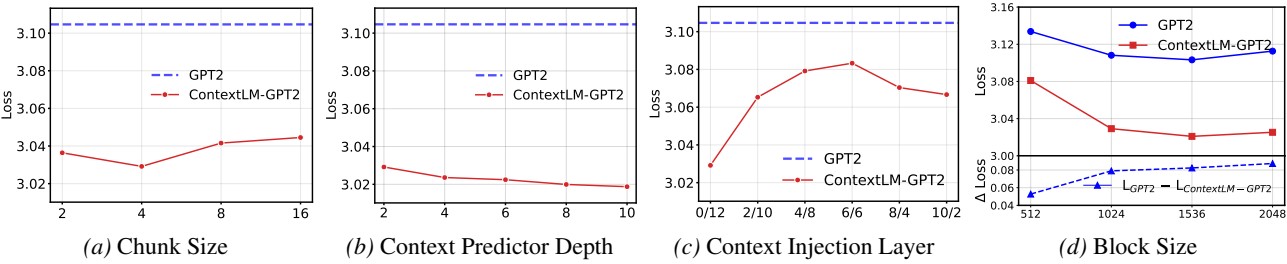

(a) Chunk Size   (b) Context Predictor Depth   (c) Context Injection Layer   (d) Block Size

Figure 4. Validation loss on OpenWebText for baseline GPT2-Base and ContextLM-GPT2-Base across chunk sizes, Context Predictor depths, context injection layer, and block size.

ContextLM operates on chunked representations, the effective sequence length is reduced by a factor of $w$, allowing more efficient context modeling of long-range context. As block size increases, the $\Delta$Loss between ContextLM-GPT2 and GPT-2 widens, indicating that context-level supervision provides increasing benefits as the context length grows. Similar trends are observed on LongBench (Bai et al., 2024) (Appendix A.6), supporting that the improvements extend beyond validation loss to evaluations involving longer input contexts.

### 4.2. Comparison with Multi-Token Prediction

To evaluate the benefit of context-level modeling over extending the token prediction horizon, we compare ContextLM with standard multi-token prediction (MTP). Following official implementation settings (Gloeckle et al., 2024), we configure MTP to predict 4 future tokens, matching the ContextLM chunk size $w = 4$. Experimental results on the GPT2-XL scale (Table 5) show that ContextLM significantly outperforms both NTP and MTP in terms of perplexity and downstream task performance. These results suggest that predicting higher-level contextual representations provides a more informative supervision signal than extending the prediction horizon through multi-token prediction.

### 4.3. Visualizing Attention Distribution

To qualitatively understand ContextLM's behavior, we visualize a representative example in Figure 5. Compared to baseline, ContextLM allocates attention more selectively toward tokens that are salient for maintaining coherence across broader context. In particular, ContextLM places substantially higher attention on the anaphoric reference "*this*" and the preceding technical concept "*battery*," indicating increased reliance on chunk-level contextual information when predicting local tokens. A similar effect is observed for the reporting phrase "*analysts said*," where ContextLM assigns moderately higher attention to the surrounding these tokens (approximately a 16% increase). Overall, this pattern suggests that by incorporating predictive context embeddings, ContextLM enhances its ability to capture both token-level dependencies and high-level contextual relationships.

## 5. Related Work

### 5.1. Modeling the semantic level above tokens

Recent studies have explored multi-level architectures that extend beyond token-level next-token prediction (NTP) to explicitly model higher-level semantics. Block Transformer (Ho et al., 2024) introduces a global-to-local modeling strategy that accelerates inference while maintaining

*Table 5.* Comparison against Multi-Token Prediction (MTP) at the GPT2-XL scale trained on OpenWebText.

| Model | PPL ↓ | Lambada OpenAI | ARC-E | Lambada Standard | ARC-C | Wino Grande | PIQA | Hella-Swag | SciQ | RACE | Avg Acc ↑ |
|---|---|---|---|---|---|---|---|---|---|---|---|
| GPT2-XL-NTP | 15.25 | 38.7 | 48.1 | 29.2 | 20.6 | 50.8 | 65.2 | 31.0 | 77.3 | 29.2 | 43.3 |
| GPT2-XL-MTP | 16.72 | 36.7 | 49.1 | 25.4 | 19.6 | 49.3 | 64.1 | 30.3 | 77.4 | 30.5 | 42.5 |
| **ContextLM-XL** | **14.60 / ↓ 0.65** | **41.9** | **49.9** | **32.3** | **21.6** | **51.9** | **66.1** | **32.3** | **77.7** | **31.2** | **45.0 / ↑ 1.7** |

**Input Text:**

Apple unveiled its latest iPhone model on Tuesday. The new device features a revolutionary graphene battery. This technology allows for significantly faster charging times and longer overall battery life, analysts said.

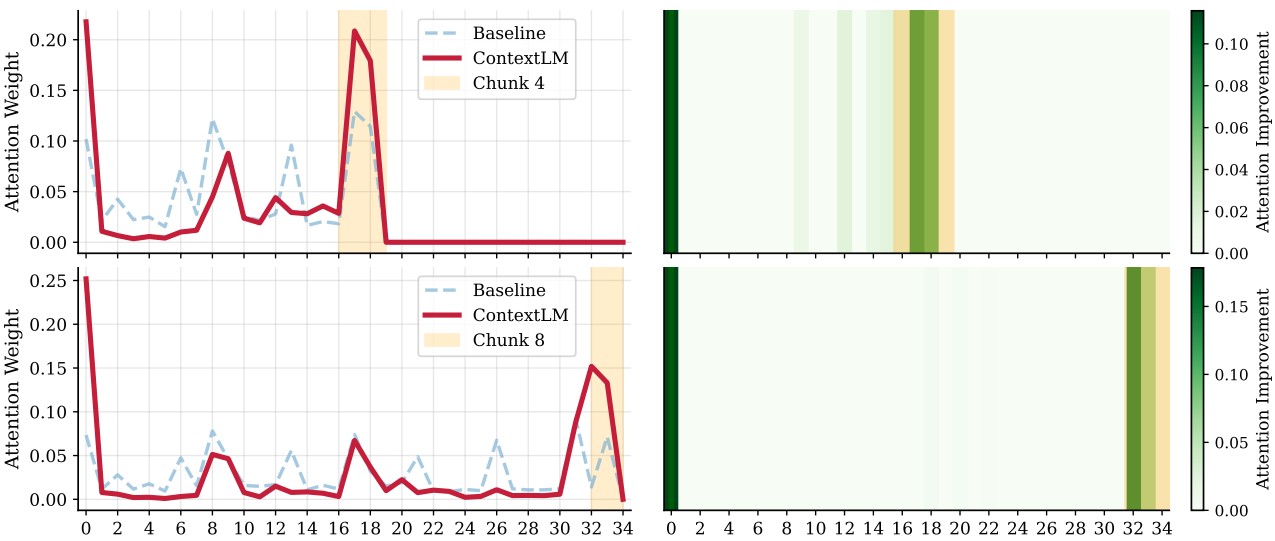

*Figure 5.* Attention weight analysis for the input text. ContextLM-GPT2-XL (red) shows a significant increase in attention over the baseline GPT2-XL (blue) towards the technical concept in Chunk 4 ("*battery. This technology*") and the contextual framing in Chunk 8 ("*analysts said.*"), indicating improved contextual understanding.

accuracy, implicitly learning sentence-level abstractions. LCM (Barrault et al., 2024) employs diffusion models to predict conceptual semantics, but yielded limited performance gains and is constrained to a pre-defined semantic space. DLCM (Qu et al., 2025) introduces latent representations within an adaptive semantic space, while CALM (Shao et al., 2025) utilizes continuous abstract states to bridge discrete tokens and high-level contextual structures. In parallel, MTP (Gloeckle et al., 2024) and patch-based multi-token prediction (Shao et al., 2024) also move beyond strict token-by-token prediction, though their focus remains on optimizing the predictive objective rather than building hierarchical representations. ContextLM differs by making high-level context prediction an explicit component of the generative process—predicting and integrating latent context embeddings at chunk boundaries—while remaining fully compatible with the standard token-level autoregressive paradigm.

**5.2. Modeling the semantic level below tokens**

At a finer granularity, several approaches construct hierarchies by grouping characters or subword units into local patches. MegaByte (Yu et al., 2023) adopts a multi-scale decoder that processes sequences at both byte and token levels, primarily targeting computational efficiency. BLT (Pagnoni et al., 2025) learns patch boundaries using an entropy-based criterion, forming character-level groupings that improve modeling efficiency. H-Net (Hwang et al., 2025) dynamically adjusts patch sizes during training through an intermediate smoothing mechanism. These methods emphasize efficiency and scalability at lower granularity, whereas ContextLM targets higher-level latent representations through explicit context prediction—complementing rather than replacing token-level generation.

## 6. Conclusion

We introduce ContextLM, a hierarchical language modeling framework that learns predictive, context-level representations while remaining fully compatible with standard token-level evaluation metrics. By introducing a Context Predictor that autoregressively models higher-level context embeddings, our approach enables multi-token prediction implicitly and leverages aggregated error signals from future

tokens to capture long-range dependencies. Experiments on the GPT-2 and Pythia families, up to 1.5B parameters, show consistent improvements in perplexity and downstream performance, with gains persisting across model scales. Further analysis demonstrates that context-level supervision strengthens long-range coherence and attention allocation, highlighting next-context prediction as a scalable and efficient direction for advancing large language models.

## Acknowledgements

This work is sponsored by the National Natural Science Foundation of China (NSFC) grant (No.62576211) and the National Key Research and Development Program of China (No. 2023ZD0121402). It is also the result of a collaborative project on novel language model architectures between Shanghai Jiao Tong University (SJTU) and the Shanghai Artificial Intelligence Laboratory. The computational resources required for pretraining the models were provided by the Shanghai AI Lab.

## Impact Statement

This paper presents work whose goal is to advance the field of Machine Learning. There are many potential societal consequences of our work, none of which we feel must be specifically highlighted here.

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

# A. Appendix

## A.1. Computational Complexity and Efficiency

### A.1.1. COMPLEXITY AND MEMORY FOOTPRINT

We compare the computational complexity and memory requirements of ContextLM with a parameter-matched baseline in Table 6.

- **Computational Complexity.** The Token Decoder in ContextLM shares the same $\mathcal{O}(T^2 d)$ attention complexity as the vanilla Transformer, where $T$ is the sequence length and $d$ is the hidden dimension. The Context Predictor operates on a chunked sequence of length $K$, resulting in a complexity of $\mathcal{O}(K^2 d) \approx \mathcal{O}(\frac{1}{w^2} T^2 d)$.

- **Memory Footprint.** The memory required for token embeddings is $\mathcal{O}(Td)$. The predicted context embeddings form a sequence of length $K$, contributing $\mathcal{O}((T/w)d)$ storage.

*Table 6.* Complexity and efficiency analysis comparing GPT2-XL-PM and ContextLM-GPT2-XL on NVIDIA V100 GPUs.

| Component / Model | Theoretical Complexity | | Practical Measurements | |
|---|---|---|---|---|
| | **Training** | **Inference** | **Train (tok/s)** $\uparrow$ | **Infer (ms/tok)** $\downarrow$ |
| *Theoretical Components* | | | | |
| Token Decoder | $\mathcal{O}(T^2 d)$ | $\mathcal{O}(Td)$ | – | – |
| Context Predictor | $\mathcal{O}((T/w)^2 d)$ | $\mathcal{O}((T/w)d)$ | – | – |
| *Practical Models* | | | | |
| GPT2-XL-PM | $\mathcal{O}(T^2 d)$ | $\mathcal{O}(Td)$ | 1275.20 | 23.05 |
| ContextLM-GPT2-XL | $\approx \mathcal{O}((1+1/w^2)T^2 d)$ | $\approx \mathcal{O}((1+1/w)Td)$ | **1275.71** | **22.88** |

### A.1.2. INFERENCE EFFICIENCY BREAKDOWN

Although ContextLM introduces an additional Context Predictor, the predictor operates on a shortened sequence of length approximately $T/w$, rather than the full token sequence of length $T$. Therefore, its overhead is small compared with the Token Decoder, especially at larger model scales. To quantify this, we report the component-wise inference breakdown in Table 7.

*Table 7.* Inference efficiency breakdown across model sizes.

| Model | Training Throughput (tok/s) $\uparrow$ | Inference Latency (ms/tok) $\downarrow$ | Token Decoder Time (%) | Context Predictor Time (%) |
|---|---|---|---|---|
| Base | 12196.85 | 5.01 | 62.52 | 4.55 |
| Med | 4909.99 | 8.60 | 79.14 | 2.01 |
| Large | 2463.82 | 12.61 | 87.84 | 1.25 |
| XL | 1275.71 | 22.88 | 93.60 | 0.89 |

## A.2. Training Hyperparameters on GPT-2

In our scaling law experiments, we adopt the configurations detailed in Table 8 for the GPT-2 model family. All models are trained on the OpenWebText with a maximum sequence length of 1024 tokens. Optimization is performed using AdamW ($\beta_1 = 0.9$, $\beta_2 = 0.95$) with gradient clipping at 1.0 and linear warmup over the first 1,000 steps. Learning rates are scaled according to model size following established practices.

*Table 8.* Training hyperparameters for GPT-2 family models.

| Model | $n_{\text{head}}$ | $d_{\text{model}}$ | learning rate | batch size | tokens |
|---|---|---|---|---|---|
| GPT2-Base | 12 | 768 | 1.0e-3 | 0.5M | 9B |
| GPT2-Medium | 16 | 1024 | 8.0e-4 | 0.5M | 9B |
| GPT2-Large | 20 | 1280 | 6.0e-4 | 0.5M | 9B |
| GPT2-XL | 25 | 1600 | 4.0e-4 | 0.5M | 9B |

### A.3. Training Hyperparameters on Pythia

For the Pythia model family, we detail the training hyperparameters in Table 9. Consistent with standard practices, the Pythia models are trained with a global batch size of 2M tokens for a total of 300B tokens.

*Table 9.* Training hyperparameters for Pythia family models.

| Model | $n_{\text{head}}$ | $d_{\text{model}}$ | learning rate | batch size | tokens |
|---|---|---|---|---|---|
| Pythia-70M | 8 | 512 | 1.0e-3 | 2M | 300B |
| Pythia-160M | 12 | 768 | 6.0e-4 | 2M | 300B |
| Pythia-410M | 16 | 1024 | 3.0e-4 | 2M | 300B |
| Pythia-1B | 8 | 2048 | 2.5e-4 | 2M | 300B |
| Pythia-1.4B | 16 | 2048 | 2.0e-4 | 2M | 300B |

### A.4. Training Details of The Main Result

The primary computational overhead is incurred during the pretraining phase of ContextLM-Pythia-1B and ContextLM-Pythia-1.4B on the 300B-token Pile dataset. We conduct the pretraining on a high-performance computing cluster equipped with NVIDIA A100 GPUs (80GB VRAM). The total training compute amounts to approximately 3,594 and 5,375 GPU hours for the 1B and 1.4B models, respectively.

### A.5. Downstream Task Evaluation on GPT-2

To verify that ContextLM's improvements stem from the Context Predictor's modeling capabilities rather than simply the addition of parameters, we train an aggressively configured parameter-matched (GPT2-PM) baseline. The GPT2-PM model is constructed by adding standard Transformer layers to the vanilla GPT-2 backbone, explicitly aligning its total parameter count with that of ContextLM.

As detailed in Table 10, our model achieves consistently lower perplexity than the PM baseline on all general language modeling benchmark. We further extend the evaluation to 9 representative benchmarks under both zero-shot and five-shot settings, as summarized in Table 13. ContextLM demonstrates systematic improvements across nine representative benchmarks. These gains are consistent across all model scales, with particularly pronounced improvements on reasoning-intensive tasks such as HellaSwag and PIQA.

The stable performance enhancement across both perplexity-based and task-based metrics substantiates that context-level supervision strengthens generalization capability and promotes more robust compositional understanding across datasets and model sizes.

### A.6. Evaluation on Long-Context Benchmarks

We extend our evaluation to long-context reasoning LongBench (Bai et al., 2024) benchmarks using the Pythia-1.4B backbone. To handle input sequences exceeding the model's pre-training limit of 2048 tokens, we directly apply dynamic NTK-aware RoPE interpolation. As shown in Table 11, ContextLM consistently outperforms the baseline across all LongBench task categories, achieving a +3.11 improvement in overall average accuracy.

*Table 10.* Perplexity comparisons between GPT-2, GPT2-PM and ContextLM-GPT2 across four benchmark datasets. ContextLM consistently achieves lower perplexity across all model scales.

| Model | OWT | Wikitext | Lambda OpenAI | Lambada Standard | Avg PPL ↓ |
|---|---|---|---|---|---|
| GPT2-Base | 22.38 | 45.41 | 74.06 | 301.82 | 110.92 |
| GPT2-Base-PM | 21.61 | 43.17 | 66.84 | 299.46 | 107.77 |
| **ContextLM-Base** | **20.68**↓ 1.70 | **41.45**↓ 3.96 | **55.22**↓ 18.84 | **231.41**↓ 70.41 | **87.19**↓ 23.73 |
| GPT2-Medium | 18.10 | 35.78 | 36.53 | 131.43 | 55.46 |
| GPT2-Medium-PM | 17.87 | 34.35 | 34.14 | 126.35 | 53.18 |
| **ContextLM-Medium** | **17.03**↓ 1.07 | **32.08**↓ 3.70 | **28.06**↓ 8.47 | **95.06**↓ 36.37 | **43.06**↓ 12.40 |
| GPT2-Large | 16.22 | 30.92 | 26.68 | 79.27 | 38.27 |
| GPT2-Large-PM | 16.18 | 31.04 | 24.79 | 73.12 | 36.28 |
| **ContextLM-Large** | **15.41**↓ 0.81 | **28.82**↓ 2.10 | **20.89**↓ 5.79 | **62.09**↓ 17.18 | **31.80**↓ 6.47 |
| GPT2-XL | 15.25 | 28.98 | 22.76 | 59.17 | 31.54 |
| GPT2-XL-PM | 15.14 | 28.23 | 22.08 | 60.82 | 31.57 |
| **ContextLM-XL** | **14.60**↓ 0.65 | **27.05**↓ 1.93 | **17.46**↓ 5.30 | **44.62**↓ 14.55 | **25.93**↓ 5.61 |

*Table 11.* LongBench Performance Summary (0-4k Subtasks). ContextLM demonstrates superior long-context reasoning capabilities, consistently outperforming the Pythia-1.4B baseline across diverse task categories.

| Model | Single-Doc | Multi-Doc | Summary | Synthetic | Code & Class. | Overall Avg ↑ |
|---|---|---|---|---|---|---|
| Pythia-1.4B | 13.17 | 4.49 | 13.97 | 2.32 | 33.94 | 20.21 |
| **ContextLM-Pythia-1.4B** | **13.41** | **4.92** | **20.19** | **4.39** | **40.40** | **23.32** |

## A.7. Ablation of Aggregation Strategy

We evaluate different aggregation mechanisms using Pythia-ContextLM-410M at the 30k-step checkpoint. As shown in Table 12, the performance differences across different aggregation strategies are marginal. Consequently, we select Mean Pooling as our default design, as it achieves competitive performance comparable to more complex parameterized methods without introducing any additional parameters.

*Table 12.* Ablation of Aggregation Strategies on Pythia-ContextLM-410M at the 30k-step checkpoint.

| Strategy | Loss ↓ |
|---|---|
| **Mean Pooling (Ours)** | **2.2919** |
| MLP Pooling | 2.2930 |
| Latent Attention Pooling | 2.2953 |
| Self-Attention Pooling | 2.2907 |

## A.8. Training Curve Comparison

In this section, we compare the training loss convergence of ContextLM-Pythia against the Pythia baseline. Both models are trained from scratch on the Pile dataset (300B tokens) using the same data and hyperparameters. As Figure 6 demonstrated, ContextLM-Pythia consistently maintains a lower loss compared to the baseline, proving that the context prediction objective introduces a more efficient gradient signal that sustains long-term performance gains.

*Table 13.* Downstream task accuracy across nine benchmarks for GPT-2, GPT2-PM and ContextLM-GPT2 under zero-shot and five-shot settings. ContextLM-GPT2 consistently outperforms GPT-2 and GPT2-PM across all model scales.

| Model | Lambda OpenAI | ARC-E | Lambda Standard | ARC-C | Wino Grande | PIQA | Hella-Swag | SciQ | RACE | Avg Acc ↑ |
|---|---|---|---|---|---|---|---|---|---|---|
| | | | | | *Zero-shot* | | | | | |
| GPT2-Base | 27.0 | 42.1 | 20.1 | 18.2 | 49.6 | 60.3 | 27.5 | 67.5 | 27.5 | 37.8 |
| GPT2-Base-PM | 27.3 | 41.2 | 19.8 | 18.9 | 50.3 | 60.2 | 27.6 | 69.2 | 27.2 | 38.0 |
| **ContextLM-Base** | 29.2 | 43.8 | 20.3 | 19.3 | 52.9 | 60.7 | 27.6 | 69.5 | 28.4 | **39.1 / ↑ 1.3** |
| GPT2-Medium | 33.3 | 44.1 | 23.9 | 19.3 | 50.5 | 62.9 | 29.0 | 72.7 | 29.1 | 40.5 |
| GPT2-Medium-PM | 33.9 | 45.2 | 24.9 | 19.8 | 50.3 | 62.8 | 29.1 | 74.0 | 29.3 | 41.0 |
| **ContextLM-Medium** | 36.8 | 46.7 | 26.7 | 19.5 | 50.9 | 62.9 | 29.8 | 74.2 | 29.3 | **41.9 / ↑ 1.4** |
| GPT2-Large | 36.2 | 45.9 | 26.5 | 20.5 | 48.3 | 63.6 | 30.3 | 73.5 | 28.1 | 41.4 |
| GPT2-Large-PM | 37.3 | 47.2 | 26.9 | 20.2 | 51.4 | 64.3 | 30.3 | 74.6 | 29.7 | 42.4 |
| **ContextLM-Large** | 40.7 | 47.9 | 29.9 | 21.1 | 51.3 | 65.7 | 31.2 | 76.6 | 31.0 | **43.9 / ↑ 2.5** |
| GPT2-XL | 38.7 | 48.1 | 29.2 | 20.6 | 50.8 | 65.2 | 31.0 | 77.3 | 29.2 | 43.3 |
| GPT2-XL-PM | 38.2 | 49.4 | 29.0 | 20.7 | 50.7 | 64.3 | 31.1 | 75.4 | 30.0 | 43.2 |
| **ContextLM-XL** | 41.9 | 49.9 | 32.3 | 21.4 | 51.9 | 66.1 | 32.3 | 77.7 | 31.2 | **45.0 / ↑ 1.7** |
| | | | | | *Five-shot* | | | | | |
| GPT2-Base | 18.3 | 40.8 | 17.0 | 18.6 | 51.9 | 59.8 | 27.3 | 69.0 | 26.8 | 36.6 |
| GPT2-Base-PM | 18.8 | 40.5 | 16.7 | 18.9 | 50.4 | 60.3 | 28.0 | 69.4 | 27.4 | 36.7 |
| **ContextLM-Base** | 19.4 | 42.2 | 17.9 | 18.8 | 51.7 | 60.8 | 27.8 | 72.4 | 27.2 | **37.6 / ↑ 1.0** |
| GPT2-Medium | 22.8 | 45.2 | 20.4 | 20.1 | 49.0 | 62.5 | 28.9 | 71.8 | 28.5 | 38.8 |
| GPT2-Medium-PM | 22.7 | 44.7 | 21.6 | 21.4 | 48.7 | 61.3 | 29.2 | 75.4 | 30.0 | 39.4 |
| **ContextLM-Medium** | 24.0 | 47.3 | 22.0 | 20.1 | 51.1 | 64.2 | 29.8 | 75.3 | 28.4 | **40.2 / ↑ 1.4** |
| GPT2-Large | 25.3 | 46.6 | 23.1 | 20.6 | 51.5 | 65.1 | 30.4 | 79.1 | 27.7 | 41.0 |
| GPT2-Large-PM | 25.4 | 47.0 | 24.0 | 21.8 | 51.0 | 63.9 | 30.5 | 80.1 | 28.2 | 41.3 |
| **ContextLM-Large** | 29.7 | 49.7 | 26.4 | 21.1 | 51.5 | 65.2 | 31.2 | 82.0 | 30.7 | **43.1 / ↑ 2.1** |
| GPT2-XL | 28.0 | 49.0 | 24.3 | 21.2 | 50.6 | 65.4 | 30.9 | 81.3 | 28.4 | 42.1 |
| GPT2-XL-PM | 27.5 | 50.5 | 25.1 | 21.3 | 50.9 | 64.9 | 31.3 | 80.4 | 29.3 | 42.4 |
| **ContextLM-XL** | 30.7 | 50.6 | 29.3 | 22.2 | 51.0 | 65.9 | 32.2 | 83.6 | 30.8 | **44.0 / ↑ 1.9** |

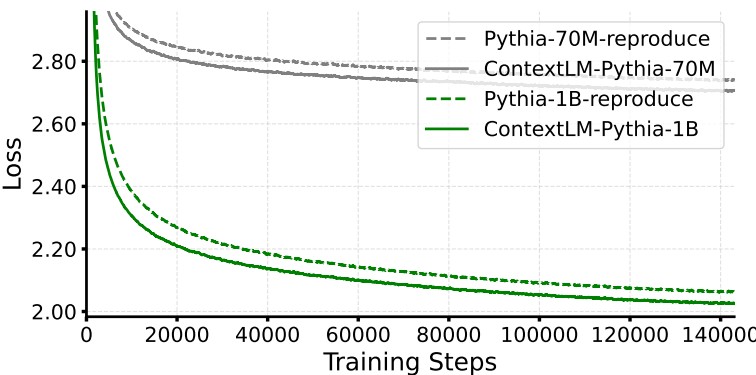

*Figure 6.* Training curve for Pythia vs ContextLM training on Pile (300B tokens). Comparison 70M and 1B baselines to control for any differences in training configurations.

## A.9. Analysis of Predicted Context Embeddings

### A.9.1. VISIBLE HISTORY INTERVENTION

To examine whether the predicted context embeddings $\hat{c}$ provide useful information during inference, we conduct an intervention experiment by restricting the decoder's access to token-level history. Specifically, we modify the self-attention mask so that each token can only attend to a prefix subset of its previous tokens, while keeping the predicted context embeddings available. We define a visible history ratio, where $1.0$ corresponds to standard full-context decoding and $0.0$ restricts the decoder to only the current token together with the predicted context embedding.

*Table 14.* Validation loss under different visible history ratios.

| Visible History Ratio | 1.0 | 0.5 | 0.1 | 0.01 | 0.0 |
|---|---|---|---|---|---|
| Baseline-PM | 2.95 | 9.01 | 9.40 | 10.23 | 11.66 |
| **ContextLM** | **2.81** | **3.48** | **3.55** | **6.30** | **11.41** |

Table 14 shows that ContextLM is substantially more robust when token-level history is removed. For example, when the visible history ratio is reduced to $0.1$, the loss of the parameter-matched baseline increases to $9.40$, while ContextLM remains at $3.55$. This indicates that the predicted context embeddings encode multi-token predictive information that can partially compensate for missing token-level context.

### A.9.2. NOISE INJECTION

We further test whether ContextLM genuinely relies on the predicted context embeddings by injecting Gaussian noise into $\hat{c}$ during inference. The injected noise is normalized to match the scale of the original context embeddings, and we gradually increase the noise ratio.

*Table 15.* Effect of injecting Gaussian noise into predicted context embeddings during inference.

| Noise Ratio | 0% | 10% | 20% | 50% | 100% |
|---|---|---|---|---|---|
| Val PPL $\downarrow$ | 20.68 | 29.57 | 40.89 | 90.73 | 276.91 |

As shown in Table 15, corrupting the predicted context embeddings leads to a sharp increase in perplexity. This behavior supports the conclusion that $\hat{c}$ is not merely a training-time regularizer; instead, it provides meaningful predictive information that directly influences generation.

## A.10. Case Studies

### A.10.1. ATTENTION TO LONG-RANGE SEMANTIC ANCHORS

To further analyze how ContextLM utilizes context-level information, we provide additional case studies on attention allocation. For each example, we identify a query token whose interpretation depends on earlier semantic anchors, and compare the total attention weight assigned to these anchor tokens by the baseline and ContextLM. As shown in Table 16, ContextLM assigns higher attention to semantically relevant long-range anchors across diverse scenarios, including biomedical, legal, geographic, medical, cybersecurity, and technology-related contexts.

*Table 16.* Additional Attention Weight Cases. A quantitative extension of the visualization in Figure 5.

| Query Token | Anchor Tokens | Baseline | ContextLM | Improvement |
|---|---|---|---|---|
| **Case 1:** In the vaccine trial, participants developed strong neutralizing antibodies. Researchers said those **antibodies** were associated with lower infection risk. | | | | |
| antibodies | vaccine, neutralizing, trial | 12.30% | 21.92% | **+9.63%** |
| **Case 2:** The court reviewed the law after a constitutional challenge from civil groups. In its final opinion, the court called the provision **unconstitutional**. | | | | |
| unconstitutional | court, law, challenge | 10.38% | 19.21% | **+8.83%** |
| **Case 3:** He sat by the river and watched the water move past the rocks. After the rain, the grass on the **bank** was still wet. | | | | |
| bank | river, water | 5.77% | 11.93% | **+6.16%** |
| **Case 4:** The patient had persistent abdominal pain and elevated white blood cell count. After reviewing the CT scan, the doctor documented a final **diagnosis**. | | | | |
| diagnosis | patient, ct, abdominal | 6.12% | 12.12% | **+6.00%** |
| **Case 5:** Attackers encrypted several hospital servers overnight and disabled scheduling systems. The IT team restored backups while investigators confirmed a **ransomware** incident. | | | | |
| ransomware | encrypted, servers, backups | 13.09% | 16.94% | **+3.85%** |
| **Case 6:** Apple unveiled a new phone with a revolutionary graphene battery. Reviewers highlighted better power density and thermal behavior. This **technology** significantly reduces charging time. | | | | |
| technology | graphene, battery | 8.67% | 10.96% | **+2.30%** |

These results suggest that ContextLM does not merely improve perplexity through additional parameters. Instead, the context-level prediction pathway encourages the model to better preserve and reuse semantically relevant information from earlier tokens, which is particularly useful when the correct interpretation of a later token depends on long-range contextual cues.

### A.10.2. QUALITATIVE LATENT SEMANTIC RETRIEVAL

We further examine whether the predicted context embeddings $\hat{c}_{t+1}$ encode human-interpretable semantic information in Table 17. To this end, we perform nearest-neighbor retrieval in the latent context embedding space. Given a predicted context embedding, we retrieve the nearest context chunks from the embedding space and decode their corresponding token chunks. If the predicted context embedding captures the correct high-level continuation, the nearest retrieved chunks should align with the context-dependent answer.

*Table 17.* **Qualitative latent semantic retrieval.** Nearest-neighbor chunks retrieved from $\hat{c}_{t+1}$ align with the correct context-dependent continuation.

**[Case 1: Business / Dynamic Entity Binding]**

**Context Prefix:** *"To ensure absolute secrecy during the merger, the acquiring firm was code-named 'Apollo' and the target firm was 'Zeus'. As the lead negotiator for the acquiring firm, she introduced herself as representing"*

| Rank | Retrieved Latent Chunk ($w = 4$) | |
|---|---|---|
| **1** | **[" Apollo", ".", " The", " talks"]** | **Correct** |
| 2 | [" Zeus", ".", " The", " talks"] | Wrong |
| 3 | [" Athens", ".", " The", " talks"] | Wrong |
| 4 | [" Google", ".", " The", " talks"] | Wrong |

**[Case 2: Science / Causal State Deduction]**

**Context Prefix:** *"The newly synthesized compound exhibits a unique property: it remains perfectly solid at room temperature but instantly vaporizes when exposed to ultraviolet light. After turning on the ultraviolet lamp in the lab, the compound"*

| Rank | Retrieved Latent Chunk ($w = 4$) | |
|---|---|---|
| **1** | **[" instantly", " turned", " into", " gas"]** | **Correct** |
| 2 | [" slowly", " melted", " into", " liquid"] | Wrong |
| 3 | [" caught", " on", " fire", " and"] | Wrong |
| 4 | [" remained", " completely", " solid", " and"] | Wrong |

**[Case 3: Procedural / Logical Rule Following]**

**Context Prefix:** *"According to the facility's strict security protocol, temporary visitors must be issued a red badge, whereas permanent employees receive a blue badge. Because Dr. Evans is a permanent employee, she was given a"*

| Rank | Retrieved Latent Chunk ($w = 4$) | |
|---|---|---|
| **1** | **[" blue", " badge", " at", " the"]** | **Correct** |
| 2 | [" green", " badge", " at", " the"] | Wrong |
| 3 | [" red", " badge", " at", " the"] | Wrong |
| 4 | [" yellow", " badge", " at", " the"] | Wrong |

These examples provide qualitative evidence that predicted context embeddings capture abstract semantic states and can encode information beyond adjacent token-level co-occurrence.

## A.11. Limitation and Future Work

### A.11.1. LIMITATION

There are two primary limitations to our current work. Firstly, limited by computational resources, we have only scaled ContextLM up to 1.5B parameters. Although the robust scaling trends observed in Figure 1 strongly suggest that our efficiency gains and performance improvements will extrapolate to larger scales, the specific behavior and efficiency of our method in the regime of large foundation models (e.g., >7B) remain to be empirically verified. Secondly, the current architecture employs a fixed chunk size (e.g., $w = 4$) for context aggregation. While computationally efficient, this fixed granularity may not perfectly align with the variable lengths of natural semantic units in diverse linguistic contexts.

### A.11.2. FUTURE WORK

In the future, we aim to move beyond fixed chunk size processing by introducing dynamic chunking, where context boundaries are determined adaptively based on entropy thresholds or linguistic cues to better capture variable-length semantic units. Another crucial direction is extending the framework to multimodal and code modeling domains and pursuing a theoretical understanding of how context-level objectives enhance gradient propagation and the learning of higher-level latent representations.

