# OpenReview forum: "Context-level Language Modeling by Learning Predictive Context Embeddings"
_ICML.cc/2026/Conference — ICML 2026 regular_

### Official Review · Reviewer_3NdF · 2026-03-09

**Soundness:** 3
**Presentation:** 3
**Significance:** 2
**Originality:** 3
**Overall Recommendation:** 4
**Confidence:** 4

**Summary:**

This paper proposes a modification of Transformer-based autoregressive language model where
an intermediate layer aggregates hidden states in a fixed-sized context window and pass the resulting
short-context embedding to succeeding tokens.
Experimental results show consistent improvement in scaling behaviors, model's perplexity, and
downstream tasks.

**Compliance With Llm Reviewing Policy:**

Affirmed.

**Key Questions For Authors:**

None.

**Limitations:**

Yes.

**Strengths And Weaknesses:**

Strength:
- Consistent improvement in scaling behaviors (Figure 1) and perplexity under controlled parameter-number (Table 1).
- The paper is well-written overall and easy to follow.
- The motivation for exploring architectural variations of LLMs is clear and relevant.

Weakness:
- As shown in Table 2, the improvement by the proposed method diminishes with model size and training data size.
  It seems to be unlikely that the improvement brings over to current, modestly sized LLMs such as 30B models.
- In abstract, the authors say they introduced "next-context prediction objective"
  and call the intermediate layers that receive and produce chunk-level representation "Context Predictor."
  However, as far as I understand, the loss used in the training is merely the token-level cross-entropy loss.
  It is not consistent with the abstract and I am not sure to what extent it is appropriate to call the intermediate layer
  ``Context Predictor'' since hidden states of a vanilla Transformer still trained to be helpful for predicting
  next several (or more) tokens directly through self-attention.

---

> ### Author Rebuttal · Authors · 2026-03-29
>
> We sincerely thank the reviewer for recognizing the clear motivation, the consistent improvements in scaling behaviors, and the overall writing quality of our paper. Below we address your valid concerns regarding scalability and terminology.
>
> > **W1**: As shown in Tab. 2, the improvement by the proposed method diminishes with model size and training data size. It seems to be unlikely that the improvement brings over to current, modestly sized LLMs such as 30B models.
>
> We agree that scaling behavior is a critical consideration. While pre-training a 30B model from scratch strictly exceeds typical academic compute budgets, our evaluation rigorously follows the recommended academic pre-training standard (e.g., 1B model + 30B tokens, as discussed in [1]). In fact, we have successfully scaled our experiments up to 1.4B parameters and 300B tokens.
>
> The observed decrease in *relative* improvement with scale is consistent with established scaling law behavior. As models become stronger, perplexity approaches lower regimes where achieving the same relative percentage reduction becomes increasingly difficult (noting that $PPL = e^{Loss}$). Crucially, as shown in Fig. 1, the absolute performance gap between ContextLM and the baseline persists stably across multiple model scales (124M to 1.5B), architecture families (GPT-2, Pythia), and datasets. This consistently maintained gap in the scaling curve suggests that the gains of context-level predictive modeling remain present as models grow larger.
>
> [1] ICLR 2025 Invited Talk: https://iclr.cc/virtual/2025/invited-talk/36784, P78
>
> > **W2**: In abstract, the authors say they introduced "next-context prediction objective" and call the intermediate layers that receive and produce chunk-level representation "Context Predictor." However, as far as I understand, the loss used in the training is merely the token-level cross-entropy loss. It is not consistent with the abstract and I am not sure to what extent it is appropriate to call the intermediate layer ``Context Predictor'' since hidden states of a vanilla Transformer still trained to be helpful for predicting next several (or more) tokens directly through self-attention.
>
> We thank the reviewer for pointing out this potential ambiguity. ContextLM does not introduce a separate auxiliary loss on context embeddings. Instead, the context embeddings are optimized *implicitly* through the standard token-level cross-entropy objective.
>
> Because each predicted context embedding influences multiple future token predictions within its chunk, its parameters receive aggregated gradient signals from multiple tokens simultaneously. This differs from a standard Transformer hidden state, which primarily receives supervision from individual token prediction steps. Since it explicitly carries the predictive information about the upcoming chunk, we term this module the "Context Predictor". In this sense, it is structurally optimized to minimize this aggregated future signal without introducing an explicit auxiliary loss, which we describe as an **intrinsic next-context prediction objective**.
>
> We agree that this terminology can be clarified further, and we will revise the abstract and introduction to more explicitly explain that the objective arises from multi-token gradient aggregation in latent context space, rather than from an additional loss term.

---

> > ### Author Rebuttal · Reviewer_3NdF · 2026-04-03
> >
> > Thank you for your comments.
> > I still believe my main concern (W1) is valid.

---

> > > ### Author Response · Authors · 2026-04-06
> > >
> > > Thank you for your reply. We believe we have addressed your concerns as thoroughly as possible in our previous response, and we are more than happy to resolve any additional questions you may have. Considering that this is our final opportunity to reply during the discussion period and no further specific points of confusion were raised, we will refine our paper during the camera-ready stage based on any further suggestions you might have.
> > >
> > > We sincerely appreciate your recognition of our work and thank you for maintaining your positive score.

---

### Official Review · Reviewer_aLdQ · 2026-03-12

**Soundness:** 2
**Presentation:** 3
**Significance:** 2
**Originality:** 3
**Overall Recommendation:** 4
**Confidence:** 4

**Summary:**

This paper proposes ContextLM, a language modeling framework that enhances the traditional Next-Token Prediction (NTP) task by predicting "context embeddings" in the latent space. Unlike methods that explicitly predict multiple future tokens, ContextLM introduces a lightweight Context Predictor to incorporate macro-semantic guidance during the generation process. Experiments conducted on GPT-2 and Pythia architectures demonstrate that this approach significantly shifts the Scaling Law Pareto frontier, achieving lower perplexity with substantially fewer parameters and a smaller computational budget.

**Compliance With Llm Reviewing Policy:**

Affirmed.

**Final Justification:**

The rebuttal has addressed my main concerns.

**Key Questions For Authors:**

1. The paper highlights a reduction in FLOPs and training tokens. However, the Context Predictor introduces additional sequential computation during the forward pass. Could the authors provide a comparison of actual training time to reach a specific perplexity baseline? Is the overhead of the prediction head compensated by the faster convergence in terms of token count?
2. One suggestion for reference: Could you provide a qualitative example demonstrating how the predicted latent vectors map back to human-readable semantic concepts and align with the actual context? Specifically, could you train a semantic vector decoder to see if its output semantics (in text form) accurately align with the semantics of the following text based on the preceding tokens? (This doesn't need to be added to the experiment, as there's already a perplexity metric; it's merely a human-readable example to enhance the paper's readability.)

**Limitations:**

yes

**Strengths And Weaknesses:**

Strengths

1. Unlike conventional Multi-Token Prediction (MTP), which operates in the discrete token space, this work models semantic spans in the latent space. This design maintains the autoregressive logic.
2. The empirical results are impressive, showing that ContextLM can reduce parameter count by approximately 39% and training data by 23% relative to baseline models, indicating strong practical utility.
3. The architecture can be integrated into existing Transformer backbones with minimal overhead, and the inference process remains fully compatible with standard NTP evaluation paradigms.

Weaknesses

The authors claim that the predicted context embeddings ($\hat{c}_{t+1}$) provide crucial guidance for token generation. However, the current experiments do not fully decouple the direct influence of preceding tokens from the guidance provided by the predicted semantics. Since the model utilizes a residual connection to inject these embeddings and the decoder retains full visibility of all previous tokens via self-attention, it is difficult to discern whether the performance gains stem from genuine macro-level planning or if the prediction head merely acts as a form of regularization that improves the underlying token encoder's representation quality.

To validate the actual information content and "steering" power of $\hat{c}_{t+1}$, I strongly suggest the authors supplement the following experiments:

1. During inference, if the decoder's self-attention to all tokens before step $t$ is blocked (e.g., via specialized masking), can the model still generate semantically coherent and relevant text using only the current token $x_t$ and the predicted context embedding $\hat{c}_{t+1}$?
2. Apply varying levels of Gaussian noise to the predicted semantic vector $\hat{c}_{t+1}$ and observe the degradation curve of generation quality. A marginal drop in performance would suggest the model is still primarily relying on the NTP path rather than the proposed semantic guidance.

---

> ### Author Rebuttal · Authors · 2026-03-29
>
> We sincerely thank the reviewer for careful reading and insightful suggestions. We conduct additional experiments to clarify the role of the predicted context embeddings and address the concerns raised.
>
> > **W1**: Isolating the effect of $\hat{c}_{t+1}$ via history masking.
>
> To explicitly isolate the contribution of the predicted context embedding, we conduct an intervention-based truncated history masking experiment. This intervention is not intended to replace token history, but to test whether predicted context embeddings make a non-trivial causal contribution under the standard autoregressive setup. During inference, we control the decoder’s access to historical tokens using a **visible history ratio**, where $1.0$ corresponds to full-context decoding and $0.0$ restricts the model to only the current token $x_t$ and the predicted context embedding $\hat{c}_{t+1}$. We report last token loss under different visible history ratios:
>
> | **Visible History Ratio**  | **1.0**  | **0.5**  | **0.1**  | **0.01** | **0.0**   |
> | ----------- | -------- | -------- | -------- | -------- | --------- |
> | Baseline-PM | 2.95     | 9.01     | 9.40     | 10.23    | 11.66     |
> | ContextLM   | **2.81** | **3.48** | **3.55** | **6.30** | **11.41** |
>
> ContextLM degrades substantially less as token history is removed, indicating that $\hat{c}_{t+1}$ provides useful predictive information during inference rather than acting purely as a training regularizer.
>
> > **W2**: Impact of noise injection on generation quality.
>
> Following the reviewer’s suggestion, we inject Gaussian noise into predicted context embeddings during inference, progressively replacing context embeddings with noise sampled from $\mathcal{N}(0,1)$ and normalized to match context embedding scale:
>
> | Noise Ratio          | 0%    | 10%             | 20%             | 50%              | 100%               |
> | -------------------- | ----- | --------------- | --------------- | ---------------- | ------------------ |
> | Val PPL $\downarrow$ | 20.68 | 29.57(**-43%**) | 40.89(**-98%**) | 90.73(**-339%**) | 276.91(**-1200%**) |
>
> Performance degrades sharply as context embeddings are corrupted, showing that the model substantially relies on the predicted context signal. Together, the masking and noise experiments indicate that context embeddings encode meaningful predictive structure that directly influences generation.
>
> > **Q1**: Actual training time to target PPL & overhead compensation
>
> We answer this from two complementary perspectives.
>
> **First, time to reach the same perplexity.** From Fig. 1(c), to reach the baseline’s final perplexity (**15.25**), the baseline requires about **$7.2×10^{19}$** training FLOPs, whereas ContextLM requires about **$5.8×10^{19}$** FLOPs, about **20% less compute**. Using the measured training throughput from our training logs, this corresponds to reducing wall-clock training time from about **9.8 hours** to about **8.1 hours**, saving roughly **1.8 hours** (**~18%**).
>
> **Second, comparison under matched model size.** We compare ContextLM-XL with a strictly matched baseline (GPT2-XL-PM), obtained by adding two standard Transformer layers to match the parameter count and additional sequential depth introduced by the Context Predictor:
>
> | Model        | Val PPL $\downarrow$           | Avg Acc $\uparrow$          | Training Throughput (tok/s)$\uparrow$ | Inference Latency (ms/tok)$\downarrow$ |
> | ------------ | ------------------------------ | --------------------------- | ------------------------------------- | -------------------------------------- |
> | GPT2-XL-PM   | 15.14                          | 43.2                        | 1275.20                               | 23.05                                  |
> | ContextLM-XL | **14.60 ($\downarrow$ 3.57%)** | **45.0 ($\uparrow$ 4.17%)** | **1275.71**                           | **22.88**                              |
>
> Because the Context Predictor operates on the shortened sequence $T/w$, its overhead is lower than applying additional Transformer layers on the full sequence. As a result, training throughput remains nearly identical, while ContextLM achieves lower final perplexity and higher downstream accuracy under the same training budget. Together, these results indicate improved **time-to-target efficiency**.
>
> > **Q2**: Qualitative example mapping latent vectors to semantic concepts.
>
> We perform qualitative latent semantic probing by mapping $\hat{c}_{t+1}$ to human-readable text via nearest-neighbor retrieval in representation space: [https://drive.google.com/file/d/1uT4rLZdsxn0i5c41DP3K67hqmmwjfWMV/view?usp=sharing]. The retrieved chunks align with the correct continuation (e.g., *Apollo* vs. *Zeus*, *blue badge* vs. *red badge*), suggesting that latent context embeddings capture predictive semantic structure conditioned on preceding context. We thank the reviewer for this helpful suggestion and will include these qualitative analyses in the appendix to improve interpretability.

---

> > ### Author Rebuttal · Reviewer_aLdQ · 2026-04-03
> >
> > I thank the authors for conducting the additional experiments I requested. The history masking and noise injection results together provide more convincing evidence that the predicted context embeddings carry genuine predictive information beyond regularization. The training time analysis also clarifies the practical overhead trade-off. The qualitative nearest-neighbor examples are a helpful addition. My concerns have been largely addressed, and I am raising my score accordingly.

---

> > > ### Author Response · Authors · 2026-04-06
> > >
> > > Thank you for your continued engagement and for raising your score! Your insightful comments are valuable in enhancing the quality of our work！Thank you again for your constructive feedback, which has significantly improved the rigor of our paper!

---

### Official Review · Reviewer_N25h · 2026-03-12

**Soundness:** 3
**Presentation:** 2
**Significance:** 3
**Originality:** 4
**Overall Recommendation:** 4
**Confidence:** 4

**Summary:**

This work proposes ContextLM, a novel hierarchical language modeling framework that augments existing next-token prediction LLMs with context-level representations. Specifically, ContextLM is comprised of three modules: a Token Encoder, a Context Predictor, and a Token Decoder. The token encoder first encodes the input tokens to obtain token embeddings. These token embeddings are then grouped into fixed-length chunks. Each chunk of tokens is mapped into a context embedding via a pooling operation. Then, the context predictor autoregressively predicts the next context embeddings. After that, the predicted context embeddings are combined with the corresponding token embeddings via element-wise addition. Finally, the token decoder performs standard next-token prediction conditioned on the combined token-context embeddings. ContextLM enables LLMs to capture higher-level semantics and long-range dependencies because of the aggregated supervision from multiple tokens. Extensive experiments on different sizes and families of LLMs have shown that the ContextLM structure can improve training efficiency and downstream performance across various tasks.

**Compliance With Llm Reviewing Policy:**

Affirmed.

**Final Justification:**

This work proposes a novel structural paradigm that is both interesting and promising even though it leaves several further works including the dynamic chunk size. I believe it is worth further exploration by the community. Therefore, I will maintain my positive score 4 as the final score for this work.

**Key Questions For Authors:**

(1) This work studies chunk size in Sec. 4.1. and finds that chunk size $w=4$ offers the best trade-off. Could the authors try to analyze why  why this particular choice works best?

(2) This work studies block size in Sec. 4.1. Could the authors clarify the relationship between the block size and the chunk size $w$ used in ContextLM? It would be helpful to understand how the improvement varies with different chunk sizes when the block size increases.

(3) In Appendix B.1, why does the inference throughput (ms/token) of ContextLM appear slightly faster than that of the baseline model? Since ContextLM introduces an additional context predictor, it would be helpful if the authors could clarify this and provide detailed breakdown for different scales of models.

(4) It is interesting to see that augmenting token-level representations with context-level information can improve LLM performance. However, why the model needs to be explicitly divided into a token encoder and token decoder with a context predictor inserted in between? Technically, we can also use a separate context augmentation module (another LLM) without restructuring the original LLM architecture.

(5) The performance gains decrease as the model size increases, which does raise some concerns about the scalability of the proposed approach. It would be helpful if the authors could provide further justification or discussion on whether the proposed method is expected to remain beneficial for larger models (e.g., 7B or beyond).

(6) In the Figure 3, we can see that ContextLM cannot enhance models' performance for coding and math instruction following tasks. Could the authors try to analyze the reason behind it? What causes this performance diminishment?

**Limitations:**

yes

**Strengths And Weaknesses:**

**Strengths:**

(1) The motivation for ContextLM is sound and meaningful. Modeling higher-level context helps LLMs better capture long-range dependencies and abstract semantic information beyond token-level patterns, which also aligns with human intuition. Therefore, it is interesting to see such a concept being extended to language modeling and demonstrated non-trivial enhancement in training efficiency and downstream tasks.

(2) The concept of ContextLM can be easily extended to existing AR LLMs. Technically, most AR LLMs can be divided into two parts: a token encoder and a token decoder. The context predictor can be initialized as additional Transformer layers. Therefore, the new structure remains compatible with existing training objectives and evaluation metrics.

(3) This new structure potentially opens a new direction for thinking about information compression if it can be scaled to larger models. For example, the context embeddings may serve as compact summaries of token chunks, and their representations or attention patterns could potentially be used to identify and compress redundant tokens while preserving high-level semantic information.

(4) The experiments conducted in this paper are comprehensive. The evaluation covers multiple model families and scales, and includes both perplexity comparisons and downstream task performance, which provides solid empirical evidence for the effectiveness of ContextLM.

**Weaknesses:**

(1) The efficiency analysis is somewhat limited. While the paper provides theoretical complexity and a single latency measurement, it would be helpful to include a more detailed breakdown of the inference cost. For example, reporting the latency or computational overhead of different components (e.g., the token decoder vs. the context predictor) and different sizes would help better understand the practical impact of introducing the additional context prediction module.

(2) The paper provides limited analysis on why context-level representations improve language modeling. While the intuition is that context embeddings capture higher-level semantics and long-range dependencies, the paper does not provide deeper analysis to explain the mechanism behind the improvement.

(3) The empirical results in Table 2 and Table 3 suggest that the performance gains decrease as model size increases. However, the paper does not discuss this trend or analyze whether the proposed method would still bring benefits for larger scales models (e.g., 7B or larger models).

(4) The attention analysis in the manuscript is also limited. The single case study in Sec. 4.3 does not provide sufficient evidence to explain why context-level representations improve model performance. It is highly recommended to include more systematic analyses, such as quantitative statistics over multiple examples or additional probing experiments.

---

> ### Author Rebuttal · Authors · 2026-03-29
>
> We thank the reviewer for the helpful questions. Below we address each concern and provide additional analysis.
>
> >**Q1/Q2**: Why does $ w=4$ work best, and how does chunk size interact with block size $T$?
>
> Our ablation results show that $w=4$ performs best, consistent with observations in related work [1,2], where similar multi-token spans are found to be effective. Intuitively, under subword tokenization (e.g., BPE), 4 tokens often correspond to roughly 2–3 words, which aligns with a natural semantic unit of human language. Furthermore, chunk size $w$ and block size $T$ are decoupled factors: $w$ controls how many tokens form one context embedding, while $T$ controls the total input length. We thus study them independently rather than treating them as a coupled design choice.
>
> > **W1/Q3**: Detailed inference efficiency breakdown
>
> Because the Context Predictor runs on a shortened sequence ($T/w$), it yields slightly faster inference than Baseline-PM operating on the full sequence ($T$). The latency breakdown is:
>
> | **Model** | Training Throughput (tok/s $\uparrow$) | Inference Latency (ms/tok $\downarrow$) | Token Decoder Time (%) | Context Predictor Time (%) |
> | --------- | -------------------------------------- | --------------------------------------- | ----------------------- | -------------------------- |
> | Base      | 12196.85                               | 5.01                                    | 62.52                   | 4.55                       |
> | Med       | 4909.99                                | 8.60                                    | 79.14                   | 2.01                       |
> | Large     | 2463.82                                | 12.61                                   | 87.84                   | 1.25                       |
> | XL        | 1275.71                                | 22.88                                   | 93.60                   | 0.89                       |
>
> > **Q4**: Why divide the model into encoder, predictor, decoder?
>
> Inspired by JEPA [3], this structural division is fundamentally motivated by the principle that learning directly in an abstract latent space is more effective than predicting fine-grained raw inputs. ContextLM predicts structure in an abstract latent space rather than only at the token level. The Token Encoder produces context embeddings, the Context Predictor models their evolution autoregressively, and the Token Decoder conditions next-token prediction on both token- and context-level representations.
>
> > **W3/Q5**:   Scalability to larger models
>
> We agree that scalability to larger models is an important open question. While pre-training 7B+ models from scratch is beyond typical academic budgets [4], our current experiments scale up to 1.4B parameters and 300B training tokens under a commonly adopted academic setting. As models become stronger, perplexity enters lower regimes where the same relative improvement becomes harder to achieve ($ PPL = e^{Loss}$). Taken together, the scaling trends in Fig. 1 and the empirical results across GPT-2 and Pythia demonstrate that the advantages of ContextLM remain robust across all evaluated scales.
>
> > **Q6**: Limited improvement on coding and math tasks
>
> We attribute this to two main factors. First, recent advances in LLM math and coding capabilities are often driven by post-training, especially reinforcement learning [5], whereas ContextLM is evaluated as a foundational pre-trained model without such enhancements. Second, coding and math are highly sensitive to exact token boundaries and local symbolic composition. Our current fixed-size chunking may not fit well with such strict boundaries. Exploring dynamic or adaptive chunking is an important direction for future work.
>
> > **W2/W4**: Why context-level representations improve performance
>
> Each predicted context embedding influences multiple future tokens and receives aggregated gradients across token spans, encouraging shared predictive structure beyond local next-token supervision. To support this, we add quantitative attention analysis across diverse semantic scenarios: [https://drive.google.com/file/d/1wmnr8RpRs_f3t7x_H1TNKt6a_u047kt6/view?usp=sharing]. ContextLM consistently assigns higher attention to relevant long-range tokens, with absolute improvements ranging from **+2.30% to +9.63%**, supporting improved modeling of long-range dependencies.
>
> [1] Ho N, et al. Block transformer: Global-to-local language modeling for fast inference. NeurIPS, 2024.
>
> [2] Shao C, et al.  Beyond Next Token Prediction: Patch-Level Training for Large Language Models. ICLR, 2025.
>
> [3] LeCun Y. A path towards autonomous machine intelligence version 0.9. 2, OpenReview, 2022.
>
> [4] ICLR Invited Talk: https://iclr.cc/virtual/2025/invited-talk/36784, P78
>
> [5] Shao Z, et al. DeepSeekMath: Pushing the Limits of Mathematical Reasoning in Open Language Models. arXiv, 2024.

---

> > ### Author Rebuttal · Reviewer_N25h · 2026-04-03
> >
> > Thanks for the clarification. I still have some concerns regarding the explanation for the limited improvement on coding and math tasks.
> >
> > While it is true that recent progress in coding and math has been largely driven by RL-based post-training, this factor alone may not fully explain the observed gap, since other tasks could also benefit from similar post-training techniques. Empirically,  the paper shows that the coding, extraction, and math performance of Pythia-ContextLM-1.4B is comparable to or even lower than that of Pythia-1.4B, whereas other tasks show clear improvements. This suggests that the second explanation is likely the dominant factor.
> >
> > If so, the use of fixed-size chunking may indeed impose limitations for tasks that require precise token-level reasoning and strict boundary alignment. I agree that exploring more adaptive chunking strategies would be an important direction for future work. One easy sanity check is that just increase the chunking size and see if the performance of coding and math tasks can increase.
> >
> > That said, I understand that a single paper cannot address all limitations. This work proposes a novel structural paradigm that is both interesting and promising, and I believe it is worth further exploration by the community. Therefore, I will maintain my positive score.

---

> > > ### Author Response · Authors · 2026-04-06
> > >
> > > Thank you for your continued support and valuable feedback! We will gladly incorporate your suggested sanity check (i.e., varying the chunk size to observe its specific impact on coding and math tasks) into the appendix. However, to fundamentally solve this problem, we will explicitly emphasize the exploration of dynamic chunking as the main direction of our future work. Thank you for your invaluable suggestions, which have made our final paper even better!

---

### Official Review · Reviewer_3ssC · 2026-03-13

**Soundness:** 3
**Presentation:** 3
**Significance:** 2
**Originality:** 3
**Overall Recommendation:** 4
**Confidence:** 4

**Summary:**

This paper proposes ContextLM, a framework that augments standard autoregressive language model pretraining with an intrinsic next-context prediction objective. Rather than operating solely at the token level, ContextLM constructs context embeddings that span multiple tokens (using fixed-size chunks with mean pooling) and adds an auxiliary objective to predict the next context embedding in a latent representation space, inspired by JEPA (Joint Embedding Predictive Architecture). The model remains fully compatible with standard token-by-token autoregressive evaluation (e.g., perplexity). Experiments are conducted on GPT-2 (Base through XL) and Pythia (70M through 1.4B) backbones, trained on up to 300B tokens from the Pile dataset. The authors show consistent improvements in perplexity across multiple benchmarks (OpenWebText, WikiText, Lambada), consistent gains on nine downstream tasks under zero-shot and five-shot settings, and improved performance on LongBench long-context reasoning tasks. A parameter-matched baseline (GPT2-PM) is included to confirm that gains are not simply from additional parameters.

**Compliance With Llm Reviewing Policy:**

Affirmed.

**Key Questions For Authors:**

1. How sensitive are the results to the chunk size w? Have you tried w = 2, 8, 16, or dynamic chunking? The fixed w=4 choice seems arbitrary. what motivated it, and how much does performance degrade for other values?

2. Can you provide a direct, controlled comparison against multi-token prediction (MTP) under matched training compute? This is the most natural baseline and its absence weakens the paper.

3. What do the context embeddings actually capture? Can you provide probing experiments, nearest-neighbor analyses, or other analyses showing that the context representations encode meaningful multi-token semantic information?

4. The ablation shows minimal difference across aggregation strategies (Table 11). Does this suggest that the specific architecture of the context predictor is less important than the presence of the auxiliary loss itself? Have you tried a simpler auxiliary loss (e.g., predicting the next token's embedding directly)?

**Limitations:**

Yes, the authors adequately discuss limitations including the fixed chunk size and limited scale. The future work section on dynamic chunking and multimodal extensions is reasonable. No significant negative societal impact concerns.

**Strengths And Weaknesses:**

Strengths:

The paper’s main strength is a clean, lightweight idea: augmenting standard Transformer training with a next-context prediction loss over multi-token context embeddings, motivated by JEPA and implemented with only a small context predictor module. It backs this up with unusually broad experiments across model scales, multiple perplexity datasets, many downstream tasks in zero-shot and few-shot settings, and long-context benchmarks. The parameter-matched control is especially important, making it more credible that gains come from the objective rather than extra parameters. Results appear consistently positive across settings, while remaining compatible with standard autoregressive perplexity evaluation, and the training curves suggest the added objective improves learning efficiency by driving lower loss throughout training.


Weaknesses:

Key limitations are that the method relies on a fixed chunk size for context aggregation, which may not align with variable-length linguistic units and is not thoroughly ablated for sensitivity or optimality. The experimental scale tops out around 1.5B parameters, leaving open whether gains persist at the larger model sizes that matter most in practice. The paper also argues it improves on multi-token prediction, but provides only limited, not fully budget-matched comparisons, and the context predictor design seems under-justified given ablations showing little difference across aggregation choices. Claims about learning higher-level or compositional representations are not backed by representation analysis or probing. Finally, downstream gains are modest relative to perplexity improvements, and the work would be better situated with comparisons against other auxiliary pretraining objectives beyond multi-token prediction.

---

> ### Author Rebuttal · Authors · 2026-03-29
>
> We thank the reviewer for recognizing our clean design and broad experiments. We address your concerns below.
>
> > **Q1**: Why $w=4$? How does performance vary with $w$?
>
> As detailed in **Sec.4.1 / Fig. 4 (a)**, we evaluate $w \in \{2, 4, 8, 16\}$ under matched compute, and find that $w=4$ provides the optimal balance between semantic aggregation and information granularity. This is also consistent with subword tokenization (e.g., BPE), where 4 tokens often correspond to a short semantic span, and with similar small multi-token settings in related work [1,2], suggesting that this granularity effectively balances information compression with predictive precision. We agree that fixed chunking is a limitation. Adaptive or dynamic chunking is an important direction, but it is also more challenging because boundary assignment introduces discrete decisions that complicate gradient-based optimization, and we leave this extension to future work.
>
> > **Q2**: Direct comparison against MTP under matched compute?
>
> We provide this comparison in **Sec.4.2 / Tab. 5** and summarize the compute trade-offs below. For GPT2-XL ($d=1600$, $V=50,257$), MTP with 4-token prediction introduces three additional LM heads, adding $≈241M$ parameters. In contrast, ContextLM ($w=4$) adds two Transformer layers, corresponding to $≈61.5M$ parameters.
>
> MTP introduces additional compute of approximately $9TdV$ from extra projection heads, whereas ContextLM adds approximately $18Td^2 + \frac{3T^2d}{4}$ from two Transformer layers operating on sequence length $T/4$.
>
> | Model             | Extra Params | Extra FLOPs         | PPL$\downarrow$                 | Avg Acc $\uparrow$         |
> | ----------------- | ------------ | ------------------- | ------------------------------- | -------------------------- |
> | MTP ($head=4$)    | $241.4M$    | $7.41\times10^{11}$ | 16.72                           | 42.5                       |
> | ContextLM ($w=4$) | $61.5M$     | $4.85\times10^{10}$ | **14.60 ($\downarrow$ 12.68%)** | **45.0 ($\uparrow$5.88%)** |
>
> Despite using fewer additional parameters and lower extra FLOPs, ContextLM yields a **12.68% relative reduction in PPL** and a **5.88% relative improvement in Avg Acc** over MTP.
>
> > **Q3**: What do context embeddings capture?
>
> We analyze the role of context embeddings via an intervention experiment that controls how much token-level history the decoder can access. Specifically, we modify the self-attention mask so that each token can attend to only a prefix subset of its preceding tokens, while keeping the predicted context embeddings available to the decoder. We define a **visible history ratio**, where 1.0 corresponds to full context, and 0.0 restricts the decoder to only the current token $x_t$ together with predicted context embedding $\hat{c}_{t+1}$. We report the last token prediction loss under different visible history ratios:
>
> | **Visible History Ratio**  | **1.0**  | **0.5**  | **0.1**  | **0.01** | **0.0**   |
> | ----------- | -------- | -------- | -------- | -------- | --------- |
> | Baseline-PM | 2.95     | 9.01     | 9.40     | 10.23    | 11.66     |
> | ContextLM   | **2.81** | **3.48** | **3.55** | **6.30** | **11.41** |
>
> ContextLM degrades substantially less as token history is removed, indicating that predicted context embeddings provide predictive information beyond local token-level signals.
>
> For interpretability, we map $\hat{c}_{t+1}$ to human-readable text via nearest-neighbor retrieval (https://drive.google.com/file/d/1uT4rLZdsxn0i5c41DP3K67hqmmwjfWMV/view?usp=sharing). Retrieved chunks align with the correct context-dependent continuation (e.g., *Apollo* vs. *Zeus*, *blue badge* vs. *red badge*), showing that the latent embeddings capture meaningful multi-token semantic structure.
>
> > **Q4**: Does weak sensitivity to aggregation imply gains mainly come from auxiliary supervision?
>
> Tab. 11 shows limited sensitivity to the aggregation operator, indicating that gains come from introducing predictive structure over multi-token representations rather than the form of the aggregation function. Similar observations have been reported in Block Transformer [1], where different aggregation strategies achieve comparable performance.
>
> Importantly, ContextLM does not introduce an explicit auxiliary loss: context embeddings are optimized implicitly through the standard token-level cross-entropy, where each predicted context embedding influences multiple future tokens, aggregating gradients across token spans. This differs from predicting a single token embedding, which remains a token-level objective. The key contribution is therefore the latent predictive structure itself, rather than the specific aggregation parameterization.
>
> [1] Ho N, et al. Block transformer: Global-to-local language modeling for fast inference. NeurIPS, 2024.
>
> [2] Shao C, et al.  Beyond Next Token Prediction: Patch-Level Training for Large Language Models. ICLR, 2025.

---

### Decision · Program_Chairs · 2026-04-30

**Decision:**

Accept (regular)

**Comment:**

This paper introduces ContextLM, a method to extend standard autoregressive language models with a context-level prediction objective. By aggregating sequences of tokens into fixed-size context embeddings, the model learns to predict subsequent high-level representations in a latent space.

Overall, while the paper presents a well-motivated and practically efficient approach, it feels somewhat incomplete regarding its depth of analysis and scaling verification. Reflecting the unanimous consensus among the reviewers, the overall assessment leans toward a Weak Accept. I concur with this recommendation and suggest acceptance, provided that the authors consider the feedback regarding representational analysis and scalability in their future work.